# Inflammatory Bowel Disease: Emerging Therapies and Future Treatment Strategies

**DOI:** 10.3390/biomedicines11082249

**Published:** 2023-08-11

**Authors:** Elisabetta Bretto, Davide Giuseppe Ribaldone, Gian Paolo Caviglia, Giorgio Maria Saracco, Elisabetta Bugianesi, Simone Frara

**Affiliations:** 1Department of Medical Sciences, University of Turin, 10126 Turin, Italy; brettoelisabetta242@gmail.com (E.B.);; 2Unit of Gastroenterology, Città della Salute e della Scienza di Torino-Molinette Hospital, 10126 Turin, Italy

**Keywords:** novel therapies, inflammatory bowel disease, Crohn’s disease, ulcerative colitis, JAK inhibitors, anti–interleukin-23 antibodies, anti-integrins, sphingosine-1-phosphate receptor modulators

## Abstract

Inflammatory bowel disease (IBD) is a term used to represent a group of chronic, relapsing inflammatory disorders of the gastrointestinal tract. Crohn’s disease (CD) and ulcerative colitis (UC) are the two major clinical forms. The global incidence and prevalence of IBD have increased over the last 2–4 decades. Despite the specific etiopathogenesis of IBD still being unknown, it is widely recognized that immunological, genetic, and environmental factors are implicated. A greater understanding of the multiple signaling pathways involved has led to the development of biologic therapies in the last two decades. Although these treatments have dramatically transformed the course of IBD, there is not a definitive cure and available therapies may cause adverse events (AEs), limiting their use, or have an inadequate effect in some patients. In this context, emerging therapies addressing new specific pathogenetic mechanisms have shown promising efficacy and safety data in early clinical trials. The purpose of this review is to highlight the available clinical trial data for these new drugs, such as more preferential JAK inhibitors, anti-IL-23 antibodies, sphingosine-1-phosphate receptor modulators, anti-integrin therapies, and other small molecules that are currently under research. We will emphasize the potential significance of these agents in shaping future treatment options.

## 1. Introduction

Inflammatory bowel diseases (IBDs) are chronic and relapsing inflammatory conditions and include Crohn’s disease (CD) and ulcerative colitis (UC) [1]. These diseases are characterized by persistent inflammation, which leads to various complications such as hospitalization, surgery, colorectal cancer, and disability, significantly impacting the quality of life of individuals affected by IBD [2]. Early and effective treatment is crucial to prevent relapses and complications and reduce steroid dependence, the need for surgery and hospital admission, and overall improve the well-being of IBD patients.

The global prevalence of IBD has been steadily increasing, placing additional strain on healthcare resources, and while its underlying causes are not fully understood, significant progress has been made in elucidating its mechanisms, leading to the development of effective treatments [3,4]. The complex pathogenesis of IBD, involving genetic predisposition and environmental factors triggering intricate inflammatory processes, offers numerous immunologic and genetic targets to explore [5,6]. 

Over the past two decades, the advent of biologic therapies has introduced the concept of treat-to-target strategies, focusing on controlling inflammation and altering the course of the disease [7,8,9,10,11,12,13]. These novel drugs resulted in improved health outcomes. However, treatment failure is observed in many patients, including primary and secondary non-responders. Additionally, biologics may carry the risk of rare but serious adverse events (SAEs), including severe infections, paradoxical autoimmune reactions, and a slight elevation in malignancy risk. Consequently, it is imperative to establish novel pharmaceutical interventions that are both highly efficient and safer, facilitating the progress of IBD treatment and enhancing clinical, endoscopic, and histological results.

Recent progress in the field of molecular biology and the comprehension of immunological pathways involved in IBD have opened up new possibilities for innovative pharmacological therapies. These include new biologics that modulate cellular signaling (e.g., anti-interleukin 12/23 agents and Janus kinase inhibitors) and hinder leukocyte trafficking (anti-integrin agents), offering improved safety profiles and pharmacokinetics, such as less frequent injections, shorter administration times, and oral therapy. 

This comprehensive review presents a thorough analysis of existing and emerging treatments for IBD, covering a wide range of promising therapies anticipated to shape the future of IBD management (Figure 1).

## 2. Materials and Methods

A thorough literature search to identify potentially relevant articles published in English within the last five years, describing phase 2/3 studies for UC and CD, was conducted. The search was performed in databases including PubMed, Google Scholar, and the ClinicalTrials.gov portal. The search incorporated the terms ‘Crohn’s disease’, ‘ulcerative colitis’, ‘clinical trial, phase 2’, ‘clinical trial, phase 3’, and ‘biologics’, either separately or combined using ‘AND’ or ‘OR’ operators. Case reports, case series, and phase 1 and phase 4 studies were excluded. Moreover, a hand-search of original articles and abstracts from recent major meetings investigating emerging biologics was performed to review the latest results of ongoing clinical trials. The pertinent phase 2/3 trials included in the review were categorized according to the mechanism of action of the drugs.

## 3. Results

The efficacy and safety data of promising phase 2/3 novel therapies in CD and UC are summarized in Table 1 and Table 2. Clinical trials currently ongoing or under development are also mentioned.

### 3.1. IL Inhibitors

#### 3.1.1. Selective Inhibitors of IL-23

IL-23, a member of the IL-12 cytokine family, is composed of p40 and p19 subunits, with p19 being unique to IL-23. IL-23 plays a crucial role in regulating and amplifying T helper 17 cells and activating various innate immune cells, which are important in the development of chronic inflammatory disorders such as UC and CD [59,60,61,62]. Risankizumab, a monoclonal antibody designed to attach to IL-23 to block its activity, is the first FDA-approved selective IL-23p19 used to treat moderate-to-severe CD patients [63,64,65].

##### Brazikumab

Brazikumab (MEDI2070), an IgG2 monoclonal antibody derived from human sources that exhibits targeted inhibition of the p19 subunit of IL-23, has been evaluated in a double-blind, placebo-controlled phase 2a clinical trial. The efficacy and safety of this drug were investigated in 119 patients with moderate-to-severe CD who had previously failed anti-TNF treatment [14]. The trial involved a 12-week induction period where patients randomly received either 700 mg of intravenous (iv) brazikumab or placebo, followed by 210 mg of subcutaneous (sc) brazikumab every four weeks. The primary outcome, defined as a clinical response (either a 100-point decrease in CD Activity Index score (CDAI) from baseline or clinical remission with a CDAI score < 150) at week 8, was achieved by 49.2% of patients in the MEDI2070 group (*n* = 59), compared to 26.7% in the placebo group. Sustained clinical response and clinical remission at weeks 8 and 24 were observed in 42.3% and 23.1% of patients in the MEDI2070 sc 210 mg group, respectively, compared to 23.1% and 11.5% in the placebo group during the double-blind period. The most common AEs reported were headache and nasopharyngitis. Currently, a phase 2b/3 study called INTREPID is recruiting participants for further evaluation of brazikumab effectiveness and tolerance [66].

##### Guselkumab

Guselkumab is a monoclonal human IgG1 antibody that specifically inhibits p19 IL-23.

A double-blind phase 2b study (QUASAR Induction Study 1) was conducted to assess the safety and effectiveness of guselkumab over a period of 12 weeks in patients with moderate-to-severe active UC [15]. A total of 313 patients were enrolled in the study and randomly assigned to receive either placebo, guselkumab at a dosage of 200 mg iv every 4 weeks, or guselkumab at a dosage of 400 mg iv every 4 weeks. Results at week 12 showed that clinical response rates were 61.4% for the lower dose of guselkumab and 60.7% for the higher dose of guselkumab, both significantly higher than placebo (*p* < 0.001). AEs were similar between the guselkumab and placebo groups. These findings indicate that guselkumab induction therapy is more effective than placebo for the treatment of active UC.

Another recent clinical trial, the phase 2a VEGA trial, aimed to assess the safety and effectiveness of induction therapy using a combination of guselkumab and golimumab (a TNFα antagonist) compared to monotherapy with either guselkumab or golimumab in 214 adult patients with UC who had not been previously treated with TNFα antagonists [31]. Patients were randomly assigned into the following three arms: guselkumab iv 200 mg at weeks 0, 4, and 8 (*n* = 71); golimumab 200 mg sc at week 0 and then 100 mg at weeks 2, 6, and 10 (*n* = 72); or a combination of these treatment regimens (*n* = 71). After 12 weeks, the combination group showed a higher proportion of UC patients achieving clinical response (83%) compared to the monotherapy groups (guselkumab: 74%; golimumab: 61%), as well as clinical remission (combination group: 36%; guselkumab: 21%; golimumab: 22%). Moreover, the combination therapy demonstrated a significantly higher rate of endoscopic improvement compared to monotherapy with golimumab or guselkumab. There were no notable differences observed in the incidence of AEs or SAEs among the treatment groups. These findings highlight the superior efficacy of combination induction therapy with guselkumab and golimumab for achieving clinical remission in UC patients at 12 weeks.

GALAXI 1 was a phase 2, double-blind, placebo-controlled, multicenter study evaluating the efficacy and safety of guselkumab in patients with moderately to severely active CD with inadequate response/intolerance to conventional therapies and/or biologics [16,17]. GALAXI employed a treat-through design over 48 weeks. The study enrolled a total of 309 patients who were randomized in a 1:1:1:1:1 ratio to receive either guselkumab at dosages of 200 mg, 600 mg, or 1200 mg, or placebo at weeks 0, 4, and 8, or ustekinumab (the reference arm) with an iv dose of 6 mg/kg at week 0 and a sc dose of 90 mg at week 8. Clinical response, defined by a decrease in CDAI score, was significantly higher in all guselkumab groups compared to placebo. Moreover, a greater proportion of patients achieved clinical remission (CDAI < 150) in the guselkumab groups (53%) compared to placebo (16.4%). By Week 48, 248 patients in the primary efficacy analysis underwent maintenance therapy as follows: placebo non-responders received ustekinumab 6 mg/kg iv, followed by 90 mg sc every 8 weeks; placebo responders received sc placebo every 4 weeks; patients initially receiving guselkumab 200 mg iv transitioned to 100 mg sc every 8 weeks; patients initially receiving guselkumab 600 mg iv switched to 200 mg sc every 4 weeks; patients initially receiving guselkumab 1200 mg iv transitioned to 200 mg sc every 4 weeks; and patients initially receiving ustekinumab 6 mg/kg iv transitioned to 90 mg sc every 8 weeks. At week 48, high proportions of patients in the guselkumab dose groups achieved clinical remission (63.9–73%) and corticosteroid-free remission (55.7–71.4%). Guselkumab demonstrated a favorable safety profile, with similar incidences of AEs across all groups, and the most common events were headache and nasopharyngitis. Several phase 3 open-label multicenter trials are being conducted to observe the efficacy of guselkumab in adult patients with luminal and perianal CD.

Finally, the DUET-CD study (NCT05242471) is a phase 2 randomized, double-blind study conducted to assess the efficacy and safety of guselkumab and golimumab combination therapy compared to guselkumab monotherapy, golimumab monotherapy, and placebo in patients with moderate-to-severe active CD. The study is ongoing, with an estimated primary completion date in 2024.

##### Mirikizumab

Both iv and sc administration are options for mirikizumab, a humanized IgG4 monoclonal antibody that targets the p19 component of IL-23. Recently, two phase 3 trials with a randomized, double-blind, placebo-controlled design were carried out (NCT03518086 and NCT03524092) to assess the efficacy of mirikizumab in patients with moderately to highly active UC [32]. During the induction trial, 1281 patients were randomized in a 3:1 ratio to receive either mirikizumab (300 mg) or placebo iv every 4 weeks for 12 weeks. In the maintenance trial, 544 patients who demonstrated a positive response to mirikizumab induction therapy were randomized in a 2:1 ratio to receive either mirikizumab 200 mg or placebo sc every 4 weeks for 40 weeks. Patients who did not respond during the induction trial were given the option to receive open-label mirikizumab as an extended induction during the initial 12 weeks of the maintenance trial. In both the induction trial (week 12) and the maintenance trial (week 52), the mirikizumab group exhibited markedly higher percentages of patients attaining clinical remission compared to the placebo group (24.2% vs. 13.3%, *p* < 0.001, and 49.9% vs. 25.1%, *p* < 0.001, respectively). Out of the 1217 patients who were administered mirikizumab during the controlled and uncontrolled phases, encompassing the open-label extension and maintenance periods in both trials, 15 encountered opportunistic infections (including 6 cases of herpes zoster infection), and 8 were diagnosed with cancer (including 3 with colorectal cancer). In contrast, among the patients who received placebo in the induction trial, only one experienced a herpes zoster infection, and none were diagnosed with cancer.

In a recent phase 2 controlled trial (SERENITY), mirikizumab was evaluated in 191 patients with moderate-to-severe CD [18]. Participants were randomized to receive placebo, 200 mg, 600 mg, or 1000 mg of mirikizumab iv every 4 weeks. Two-thirds of patients had prior exposure to anti-tumor necrosis factor (anti-TNF) therapy, and nearly half of the patients had previously experienced treatment failure with anti-TNF agents. At week 12, mirikizumab demonstrated a significantly higher endoscopic response compared to placebo in all treatment groups (200 mg: 25.8%, *p* = 0.079; 600 mg: 37.5%, *p* = 0.003; 1000 mg: 43.8%, *p* < 0.001; placebo: 10.9%). Patients who attained a 1-point enhancement in SES-CD at week 12 underwent re-randomization for either continuing iv treatment or switching to 300 mg of mirikizumab administered sc every 4 weeks. The endoscopic response at week 52 was 58.5% (24/41) in the iv group and 58.7% (27/46) in the sc group. The reported frequencies of AEs in the mirikizumab groups were similar to the placebo group. Mirikizumab demonstrated the achievement and maintenance of histologic response and remission over a period of 52 weeks. Ongoing phase 3 studies are evaluating the efficacy of mirikizumab in adults and pediatric patients with CD.

#### 3.1.2. Selective Inhibitors of Interleukin (IL)-36

The IL-36 cytokines are members of the broader IL-1 cytokine family. All IL-36 cytokines bind to the IL-36 receptor (IL-36 R).

In the past few years, many studies have highlighted the role of IL-36 R signaling in chronic inflammatory conditions including CD and UC [67]. The proinflammatory role of IL-36R and its participation in intestinal fibrosis and tissue remodeling could be a therapeutic target.

##### Spesolimab

Spesolimab, a novel humanized monoclonal immunoglobulin G1 antibody that specifically inhibits IL-36R signaling, has been evaluated in three phase 2/2a clinical trials involving patients with moderate-to-severe UC to define its safety and efficacy in inducing mucosal healing. Although spesolimab was generally well tolerated by UC patients, the efficacy endpoints were not met [33,34,68,69].

Presently, phase 2 clinical trials are being conducted to test the use of spesolimab in individuals with fistulizing CD [19,20].

#### 3.1.3. Selective Inhibitors of IL-6 Trans-Signaling

IL-6 engages in dual signaling pathways: the classical signaling route, which entails binding to its membrane-bound IL-6 receptor (IL-6 R) along with gp130 on immune and intestinal cells, and the trans-signaling pathway, in which soluble IL-6/IL-6 R complexes activate cells expressing gp130 alone [70,71]. While the classic signaling pathway is important for pathogen defense, the trans-signaling pathway is implicated in chronic inflammation. Anti-IL-6 agents have shown potential benefits in inflammatory disorders, but safety concerns, such as gastrointestinal perforations, have been raised based on previous studies with tocilizumab (a monoclonal antibody against IL-6R) in rheumatoid arthritis patients [72].

##### Olamkicept

Olamkicept (sgp130Fc) is a monoclonal antibody that selectively targets IL-6 trans-signaling, potentially providing safety benefits when compared to pan-IL-6 inhibitors. In a 12-week, open-label, phase 2a study involving 16 IBD patients (including 9 patients with UC and 7 patients with CD), olamkicept was administered iv at a dose of 600 mg every 2 weeks [21]. Clinical response and remission were evaluated after 12 weeks. Clinical remission, defined as Mayo score ≤ 2, bleeding score 0, and endoscopy ≤ 1 for UC and CDAI < 150 for CD, was achieved in 19% of patients (3/16; 2/9 UC and 1/7 CD, respectively). Clinical response, defined as a reduction in Mayo score of ≥3 points and bleeding score ≤ 1 for UC or a reduction of CDAI > 100 for CD, was observed in 44% of patients (7/16, UC: 5/9 UC and CD: 2/7, respectively). Additionally, endoscopic remission and response were assessed. Endoscopic remission, characterized by a subscore of 0 or 1 on the Mayo endoscopic component for UC or an SES-CD ≤ 4 with no ulcers for CD, was achieved by the three patients who also reached clinical remission. Endoscopic response, defined as a 1-point reduction in Mayo score for UC or a 50% reduction in SES-CD for CD, was observed in 37.5% of patients (6/16, UC: 5/9 and CD: 1/7, respectively). The most frequently reported adverse events included seasonal upper respiratory tract infections (e.g., laryngitis, rhinitis), recurrence of herpes labialis, and skin and subcutaneous disorders such as eczema or erythema. Importantly, no incidents of gastrointestinal perforation were documented during this exploratory study. To further investigate whether gp130 trans-signaling blockade causes any immune suppression in humans, as suggested by animal experimentation, a larger placebo-controlled clinical study is currently underway (NCT03235752).

Moreover, in a phase 2 placebo-controlled trial (NCT03235752) investigating the use of olamkicept in active UC, biweekly induction therapy with a 600 mg dose of olamkicept exhibited clinical effectiveness and promoted mucosal healing in patients with active UC [35]. The treatment demonstrated a favorable safety profile, as observed in the study published in abstract form. These findings support the potential development of olamkicept in IBD patients.

### 3.2. TNF Inhibitors

Biological agents using TNF as a target are a consolidated therapy nowadays, with infliximab, adalimumab, and golimumab authorized by the EMA [10,36,73].

#### 3.2.1. OPRX-106

OPRX-106, an orally administered recombinant TNF fusion protein (TNFR) produced in BY2 (lyophilized Nicotiana tabacum) plant cells, was tested in a phase 2a open-label clinical trial involving 25 patients with active mild-to-moderate UC [22]. The enrolled patients were randomly assigned to receive either 2 or 8 mg of OPRX-106 once daily for a duration of 8 weeks. Clinical response and clinical remission based on the Mayo Score, were observed in 67% and 28% of patients, respectively. OPRX-106 demonstrated good tolerability, with no reported serious AEs. Further research is necessary to thoroughly evaluate the efficacy and safety of this medication.

#### 3.2.2. V565

Oral V565 is a new type of antibody, specifically targeting TNFα, that has been engineered to withstand degradation by intestinal proteases. A recently concluded phase 2 study (NCT02976129) used a double-blind, placebo-controlled, parallel-group design to assess the effectiveness of V565 in treating patients with active CD over a period of 6 weeks [74]. The study involved 125 patients with diseases affecting the ileum or colon, who were randomly assigned in a ratio of 2:1 to receive either V565 or a placebo three times a day. They were then monitored for a duration of 28 days. The main objective was to assess the clinical response, which was defined as a reduction of at least 70% in the CDAI score along with a decrease of more than 40% in inflammatory markers (such as protein C-reactive or fecal calprotectin) by day 42. The rate of clinical response did not show a significant difference between the two groups, with 35.4% in the V565 group and 37.2% in the placebo group. However, the treatment group exhibited higher rates of improvement in endoscopic findings (56.3% compared to 30.0% in the placebo group). The frequency of SAEs was comparable in both cohorts, and there were no reported fatalities.

### 3.3. Anti-Adhesion

In recent years, integrin receptors have emerged as viable targets for novel therapies aimed at treating patients with IBD. Integrins comprise two subunits, α and β, and are present on specific B and T lymphocytes. Those associated with cell migration into gastrointestinal tissue are α2β2, α4β1, and α4β7. By antagonizing these receptors, lymphocyte migration into the gastrointestinal mucosa during the inflammatory process is inhibited. Vedolizumab, the pioneer of this receptor family, received approval from the European Medicines Agency (EMA) in 2020 for the treatment of adult patients with moderate-to-severe UC or CD [12,37].

#### 3.3.1. Abrilumab

Abrilumab, an α4β7 antibody, was tested in a phase 2b double-blind placebo-controlled study involving patients with moderate-to-severe UC who were unresponsive to conventional therapies [38]. The trial comprised 354 patients who were randomized to receive sc abrilumab (7, 21, or 70 mg) on day 1, weeks 2 and 4, and every 4 weeks; abrilumab 210 mg on day 1; or placebo. At week 8, the primary endpoint of clinical remission (total Mayo Score ≤ 2 points, no individual subscore > 1 point) was attained by 4% (placebo), 13% (abrilumab 70 mg), and 12% (abrilumab 210 mg) of patients. Moreover, clinical response and mucosal healing rates with these dosages were significantly higher compared to placebo. After an 8-week abrilumab treatment, patients with moderate-to-severe UC exhibited remission, clinical response, and mucosal healing.

Similar positive results were observed in a smaller randomized, double-blind, placebo-controlled study in Japan, where sc abrilumab was administered to 45 moderate-to-severe UC patients at 21 mg, 70 mg, or 210 mg dosages for 12 weeks followed by a 36-week open-label period in which every patient received abrilumab 210 mg every 12 weeks [39]. Clinical remission at week 8 was obtained by 12.9% of the patients in the abrilumab groups versus 0% in the placebo group. No serious AEs were reported.

#### 3.3.2. AJM-300

AJM-300 is a small-molecule α4-integrin antagonist that is administered orally. The safety and efficacy of this drug were investigated in a double-blind, placebo-controlled phase 2a clinical trial conducted by Yoshimura et al. in 2015 in 102 patients with active UC [40]. Participants were randomly assigned to receive either AJM-300 (960 mg) or placebo three times daily for 8 weeks. The results showed that the patients who received AJM-300 had significantly higher rates of clinical response and remission at week 9 compared to the placebo group (62.7% and 23.5% vs. 25.5% and 3.9%, respectively). No serious AEs were observed during the administration of AJM-300.

Subsequently, in 2022, a phase 3 trial was conducted in patients with moderately active UC (*n* = 203) [23]. The study consisted of a treatment phase where patients received AJM-300 (960 mg) or placebo three times daily for 8 weeks, followed by an open-label re-treatment phase where drug administration was continued for up to 24 weeks if endoscopic remission was not achieved or rectal bleeding persisted. By week 8, a clinical response was observed in 45% of patients in the AJM-300 group and 21% of patients in the placebo group (*p* = 0.00028). Clinical response was defined as a decrease of 30% or more in the Mayo Score, accompanied by a decrease of 1 or more in the rectal bleeding score, or a rectal bleeding subscore of 1 or less, in addition to an endoscopic subscore of 1 or less. There was no statistically significant difference in the incidence of AEs between the two groups.

AJM-300 was well tolerated and showed superiority over placebo in inducing a clinical response in patients with moderately active UC.

A similar phase 3b trial was conducted on moderate-to-severe CD patients who had failed prior therapy with infliximab [41]. The trial consisted of a 6-week open-label induction phase followed by a 20-week double-blind maintenance phase. All 539 enrolled patients received open-label AJM-300 400 mg at weeks 0, 2, and 4. Responders (CDAI decrease ≥ 100 points from baseline [CDAI-100]) at week 6 were then randomized to a 20-week double-blind maintenance phase with AJM-300 400 mg either every 2 or every 4 weeks. At week 6, 62.0% of patients achieved clinical response and 39.3% of patients achieved clinical remission. Among the 329 patients randomized to receive maintenance therapy, AJM-300 400 mg every 4 weeks demonstrated similar efficacy to dosing every 2 weeks for the maintenance of response and remission.

#### 3.3.3. Ontamalimab

Ontamalimab (PF-00547659) is a human IgG2κ anti-MAdCAM-1 monoclonal antibody that specifically hinders the binding of α4β7 integrin to the human mucosal addressin cell adhesion molecule-1 (MAdCAM-1) ligand, consequently diminishing lymphocyte migration to the intestinal tract. To evaluate its effectiveness and safety, a multicenter phase 2, randomized, double-blind, placebo-controlled clinical trial (TURANDOT) was carried out in 587 patients with moderate-to-severe UC who had either experienced treatment failure or were intolerant to at least one conventional therapy [24]. Patients were stratified based on previous anti-TNFα treatment and randomly assigned to three different groups: 7.5 mg, 22.5 mg, and 75 mg of sc ontamalimab administered at baseline and then every 4 weeks. Remission rates at week 12, defined as a Mayo Score ≤ 2 with no individual subscore > 1 and no rectal bleeding subscore, were significantly higher in the treatment groups compared to the placebo group. The greatest effects were observed with sc ontamalimab administered at doses of 22.5 mg and 75 mg. Furthermore, ontamalimab demonstrated good tolerability and a favorable safety profile.

Two different trials were conducted on CD patients. The OPERA I study was a phase 2 trial conducted in a double-blind, placebo-controlled manner to assess the efficacy of three different doses (22.5 mg, 75 mg, or 225 mg) of ontamalimab in 265 patients with CD [25]. The participants received sc injections of either a placebo or ontamalimab at weeks 0, 4, and 8, and were followed up for a period of 12 weeks. Although a higher proportion of patients in the ontamalimab groups achieved a clinical response compared to the placebo group at weeks 8 and 12, the differences were not statistically significant. Likewise, a higher percentage of patients in the ontamalimab groups attained CDAI-100 response and clinical remission, but these differences did not reach statistical significance. The safety profiles were equivalent across all treatment groups. The OPERA II study is an extension of the phase 2 trial, spanning 72 weeks, which aims to assess the long-term safety and effectiveness of ontamalimab in patients with CD who exhibited a positive clinical response in either the OPERA I or TOSCA study, a trial to evaluate the incidence of progressive multifocal leukoencephalopathy, a brain infection associated with anti-integrin drugs, in CD patients treated with such drugs [75]. In the study, 268 patients were administered sc injections of 75 mg ontamalimab every 4 weeks, and 157 patients underwent dose escalation. The prevalent AEs were flare-ups of CD, and two deaths unrelated to the drug were reported during the course of the study. The long-term efficacy showed that 37.3% of patients maintained clinical remission at week 72, and 65.6% of patients not in remission at baseline achieved remission. Ontamalimab demonstrated greater effectiveness than placebo, although the difference was not statistically significant, with a favorable long-term safety profile. Unfortunately, the clinical trial program for ontamalimab in IBD was discontinued, likely due to economic reasons.

### 3.4. SP1R Modulators

Sphingosine-1 phosphate (S1P) is a highly bioactive molecule derived from cell membranes that primarily functions by activating five G protein-coupled receptors on the cell surface (S1P1-S1P5). Upon activation, this process triggers a series of signaling events that govern a wide range of biological processes, such as lymphocyte migration, endothelial cell permeability, angiogenesis, and cell differentiation, proliferation, survival, and apoptosis. The S1P pathway has been associated with disorders related to inflammation in the gastrointestinal tract. Its modulation may prevent lymphocyte migration into the gut, which is known to be a critical factor in chronic inflammation, leading to decreased inflammation and tissue damage [42].

#### 3.4.1. Ozanimod

Ozanimod is a selective modulator of S1P receptors, specifically targeting the S1P1 and S1P5 receptors, which are found on endothelial cells and oligodendrocytes, respectively. A phase 3 multicenter, double-blind, placebo-controlled study (TRUE NORTH) was conducted to investigate ozanimod as an induction and maintenance therapy for moderate-to-severe UC [26]. A total of 645 patients were enrolled in the study and randomly assigned to receive either 1 mg per day of ozanimod or placebo. Following a 10-week induction therapy, 18.4% of individuals in the ozanimod group achieved clinical remission, while only 6.0% in the placebo group did (*p* = 0.001). Among responders to the induction therapy, 37.0% sustained clinical remission after 52 weeks, in contrast to 18.5% in the placebo group (*p* < 0.001). The most common AEs in the ozanimod group were elevated liver aminotransferase levels and lymphopenia. Based on the positive results, ozanimod was approved for the treatment of UC.

When it comes to the use of ozanimod in CD patients, a phase 2, uncontrolled, multicenter trial in adults with moderately to severely active CD was conducted (STEPSTONE) [76]. Every patient underwent a 7-day gradual increase in dosage (with 4 days on ozanimod 0.25 mg daily, followed by 3 days at 0.5 mg daily). Subsequently, they received ozanimod 1.0 mg daily for an additional 11 weeks, comprising a 12-week induction period, followed by a 100-week extension phase. Out of the 69 enrolled patients, 23.2% experienced endoscopic response at week 12. Clinical remission (CDAI < 150 points) was shown in 39.1% of patients while clinical response (CDAI decrease from baseline ≥ 100) was shown in 56.5% of patients. The most commonly reported SAEs were CD exacerbation and abdominal abscess, occurring in 9% and 3% of patients, respectively.

The ongoing YELLOWSTONE phase 3 initiative comprises randomized, double-blind, placebo-controlled induction (NCT03440372 and NCT03440385) and maintenance (NCT03464097) trials, along with an open-label extension (OLE) study (NCT03467958), with the goal of evaluating the safety and effectiveness of ozanimod in individuals with moderately to severely active CD [43].

#### 3.4.2. Etrasimod

Etrasimod is a selective oral agonist for the S1P1, S1P4, and S1P5 receptors. Recently, two independent randomized, multicenter, double-blind, placebo-controlled phase 3 trials (ELEVATE UC 12 and ELEVATE UC 52), were conducted to assess the safety and efficacy of etrasimod in adult patients with moderately to severely active UC [44]. ELEVATE UC 12 and ELEVATE UC 52, randomly assigned 354 and 433 patients, respectively, to once-daily oral etrasimod 2 mg or placebo. ELEVATE UC 12 independently assessed induction at week 12, whereas ELEVATE UC 52 comprised a 12-week induction period followed by a 40-week maintenance period with a treat-through design. In the ELEVATE UC 12 trial, the etrasimod group had a higher rate of clinical remission at the end of the 12-week induction period compared to the placebo group (25% vs. 15%, *p* = 0.026). The ELEVATE UC 52 trial showed that at both week 12 and week 52, a significantly higher percentage of patients in the etrasimod group achieved clinical remission compared to the placebo group (27% vs. 7%, *p* < 0.0001 and 32% vs. 7%, *p* < 0.0001, respectively). A favorable safety profile was observed for the molecule in both clinical trials. This drug demonstrated safety and efficacy as an induction and maintenance therapy for patients with moderate-to-severe UC.

#### 3.4.3. CBP-307

CBP-307 is an oral small molecule designed to target the S1P1 receptor. In a multicenter randomized, double-blind, placebo-controlled phase 2 study involving 145 patients with moderate-to-severe UC, this molecule was tested in two active dose arms (CBP-307 0.1 mg and CBP-307 0.2 mg) and a placebo arm [45]. At week 12, CBP-307 0.2 mg did not significantly reduce the adapted Mayo Score, but a higher proportion of patients achieved clinical remission compared to the placebo group (28.3% vs. 9.6%, *p* = 0.016). An exploratory analysis confirmed the pharmacodynamic activity of CBP-307, as it led to a reduction in lymphocyte counts. Safety results demonstrated that CBP-307 was generally well tolerated, with a similar occurrence of grade 3 or higher AEs compared to the placebo. Based on these findings, further clinical development of CBP-307 in UC is warranted.

#### 3.4.4. KRP203

KRP203 is a potent oral agonist of the S1P1 receptor. A phase 2 multicenter, double-blind, placebo-controlled study was conducted to assess its efficacy, safety, and tolerability in patients with moderately active 5-aminosalicylate-refractory UC [27]. Patients were randomly assigned to receive 1.2 mg KRP203 or placebo daily for 8 weeks. Clinical remission (partial Mayo Score of 0–1 and modified Baron Score of 0–1 with a rectal bleeding subscore of 0) was achieved by 14% of patients in the KRP203 group compared to 0% in the placebo group. The most frequent AEs were gastrointestinal disorders and headache, and their occurrence was similar between the treatment groups. Although the study did not meet the minimum clinically relevant threshold for efficacy, the results suggest that KRP203 treatment may be more effective than placebo, and further research is necessary.

#### 3.4.5. Amiselimod

Amiselimod, an orally administered selective S1P1 receptor modulator, was evaluated in a phase 2a multicenter, randomized, double-blind, parallel-group, placebo-controlled study on patients with moderate-to-severe CD [77]. The primary endpoint of achieving a clinical response (defined as a CDAI score of 100) at week 12 did not demonstrate a significant difference between the amiselimod 0.4 mg group and the placebo group (48.7% vs. 54.1%, respectively) among the 180 patients included in the study. Overall, amiselimod 0.4 mg was well tolerated, indicating a favorable safety profile, with 71.8% of patients completing the treatment period. Seven participants had SAEs and four discontinued treatment in the amiselimod group. Further research is needed to explore the potential of amiselimod in CD.

### 3.5. JAK Inhibitors

In recent years, there have been numerous studies focusing on the JAK-STAT pathway that have offered new therapeutic strategies to improve the treatment of IBD [46,78,79]. JAK, a family of intracellular tyrosine kinases comprising JAK1, JAK2, JAK3, and tyrosine kinase 2 (TYK2), plays a role in transmitting cytokine-mediated signals through the STAT pathway. These kinases are activated by various cytokine receptors, resulting in inflammation through T-cell proliferation and differentiation, and B-cell activation. In the context of IBD, IL-6, IL-12, and IL-23 are key drivers of disease activity, and their activation occurs via the JAK-STAT pathway. By blocking the activation of this pathway, the activity of several chemokines involved in mediating inflammation is halted.

#### 3.5.1. Izencitinib

Izencitinib (TD-1473), an oral gut-selective pan-JAK inhibitor, was investigated in a phase 2b clinical trial involving 239 patients with UC [28]. The primary objective, a reduction in total Mayo Score, and the main secondary endpoint of clinical remission were not achieved at week 8 compared to placebo. However, a slight dose-dependent increase in clinical response measured using the adapted Mayo Score was observed. Izencitinib demonstrated good tolerability across all doses administered.

In the context of CD, a multicenter, randomized, double-blind, phase 2 study was conducted to evaluate the safety and efficacy of izencitinib in 304 patients with moderate-to-severe CD who were corticosteroid-dependent or had shown resistance to conventional therapies [47]. Patients were randomized in a 2:3:3 ratio to receive placebo or izencitinib at 80 or 200 mg once daily for 12 weeks. Both doses of izencitinib were well tolerated without any new safety concerns; however, after 12 weeks of treatment, no statistically significant reduction in CDAI or endoscopic severity were observed.

#### 3.5.2. Ivarmacitinib

Ivarmacitinib (SHR0302), a selective JAK 1 inhibitor, was evaluated in the AMBER2 phase 2 multicenter, double-blind, placebo-controlled trial involving 146 patients with moderate-to-severe active UC [80]. Patients were randomly assigned to receive oral ivarmacitinib at different doses (8 mg once daily, 4 mg twice daily, 4 mg once daily) or placebo for 8 weeks, followed by an 8-week extension period. The 8 mg once-daily and 4 mg twice-daily dosing regimens showed significantly higher rates of clinical response and clinical remission at week 8 compared to placebo, with no notable differences in AEs. These promising results have led to the initiation of a phase 3 study to further investigate the efficacy of ivarmacitinib in UC [29].

In parallel, a phase 2 randomized, double-blind, placebo-controlled, multicenter study to investigate ivarmacitinib safety and efficacy in 112 patients with moderate-to-severe active CD (NCT03677648) has been completed [48]. The trial followed a (12 + 12)-week design and patients who completed the initial 12-week treatment entered a blind active-arm extension phase for an additional 12 weeks. The primary endpoint evaluated was clinical remission at week 12, defined as a CDAI score < 150. Results from this study are currently pending.

#### 3.5.3. Peficitinib

Peficitinib is an orally administered JAK1, JAK2, JAK3, and TYK2 (pan-JAK) inhibitor that was evaluated in a randomized, double-blind, placebo-controlled, phase 2b study for the treatment of moderate-to-severe UC [49]. Patients were randomized to receive peficitinib 25, 75, or 150 mg once daily, peficitinib 75 mg twice daily, or placebo. At week 8, a statistically significant peficitinib dose–response was not demonstrated as induction therapy, however, patients receiving peficitinib ≥ 75 mg per day or more obtained a significant clinical response, remission, and mucosal healing compared to placebo. The average rate of AEs was similar between patients receiving peficitinib and those receiving placebo (45.5% vs. 34.9%).

#### 3.5.4. Ritlecitinib and Brepocitinib

These two oral JAK inhibitors are, respectively, a JAK3 and a TYK2/JAK1 inhibitor. Their efficacy as induction therapy in active UC adult patients has been recently studied in a phase 2b randomized, double-blind study over a period of 32 weeks (VIBRATO) [50]. Among the 317 included patients, 150 received ritlecitinib (20 mg, *n* = 51; 70 mg, *n* = 49; 200 mg, *n* = 50), 142 received brepocitinib (10 mg, *n* = 48; 30 mg, *n* = 47; 60 mg, *n* = 47), and 25 received placebo for 8 weeks. Both JAK inhibitors led to a dose–response relationship, and clinical remission rates (Mayo Score ≤ 2; no Mayo subscore > 1; rectal bleeding subscore 0) at week 8 were significantly higher in the ritlecitinib 70 and 200 mg and brepocitinib 30 and 60 mg groups. The most frequently reported AEs were anemia, headache, and pharyngitis. The authors concluded that induction therapies with ritlecitinib and brepocitinib were more effective than placebo for the treatment of UC, with acceptable safety profiles so far.

#### 3.5.5. Deucravacitinib

Deucravacitinib is a TYK2 inhibitor that modulates inflammatory signals by IL-12 and IL-23, which was recently tested in a phase 2 study (LATTICE-UC) in moderately to severely active UC (modified Mayo Score of 5 to 9: endoscopic subscore ≥ 2, rectal bleeding subscore ≥ 1, stool frequency subscore ≥ 2) [51]. Patients received deucravacitinib 6 mg or placebo twice daily. At week 12, clinical remission rates and endoscopic response rates were not statistically different between the treatment and placebo groups. This study did not demonstrate the therapeutic efficacy of this molecule. A second phase 2 trial will evaluate a higher dose of deucravacitinib in patients with UC.

#### 3.5.6. OST-122

OST-122 is a treatment option for UC, CD, and potentially fibrotic lesions associated with CD, acting as a selective inhibitor of JAK 3/TYK2/ARK5. Currently, an ongoing phase 1b/2a study involving 32 patients aims to assess the safety and tolerability of this drug in individuals with moderate-to-severe UC for a duration of 28 days. Results are now pending [81].

### 3.6. Anti-TL1AR

Genome-wide association studies observed a link between a genetic variant of the tumor necrosis factor superfamily member 15 (TNFSF15) locus and IBD. TNFSF15, also called TNF-like ligand 1A (TL1A), is secreted by antigen-presenting cells and is involved with proinflammatory effects leading to chronic inflammation through the activation of T cells [82].

TL1A is upregulated in UC patients colonic mucosa and its levels are related to disease severity. Moreover, TL1A is also highly expressed in the tissue of CD patients and associated with profibrotic and severe diseases [52,83]. Therefore, TL1A inhibition could represent a therapeutic target for inflammatory diseases.

#### 3.6.1. PF-06480605

A multicenter, single-arm, open-label, phase 2a study (TUSCANY) was conducted to assess the safety, tolerability, and efficacy of the TL1A antibody PF-06480605, a fully human immunoglobulin G1 monoclonal antibody, on patients affected by moderate-to-severe UC [30]. The study included 50 patients who received 500 mg iv PF-06480605 every 2 weeks for a total of 7 doses, followed by a 3-month follow-up period. Of the 42 patients who completed the study, a statistically significant proportion (38.2%) achieved endoscopic improvement at week 14, defined as a Mayo Endoscopic Subscore of 0 or 1. Additionally, minimal histologic disease was observed after treatment (Robarts Histopathology Index ≤ 5: 33.3%). The most commonly reported AEs were UC disease exacerbation and arthralgia (six participants each). Tissue histopathology analyses further supported the efficacy of PF-06480605.

#### 3.6.2. PRA-023

PRA-023, a humanized monoclonal antibody targeting TL1A, was evaluated for safety and efficacy in a recent phase 2a open-label study involving 55 patients with moderate-to-severe CD [53]. The study population was characterized by a significant proportion of patients with previous exposure to biologic therapies (70.9%) and a mean duration of disease of 10.3 (9.3) years. IV administration of PRA-023 at a dose of 1000 mg on day 1, followed by 500 mg at Weeks 2, 6, and 10, was performed. The study results, presented at the 18th Congress of the European Crohn’s and Colitis Organisation, showed significantly higher rates of endoscopic response (reduction in SES-CD score of ≥50%) and clinical remission in patients receiving PRA-023 compared to placebo (26% vs. 12% and 49% vs. 16%, respectively, with *p*-values < 0.001). PRA-023 was well-tolerated, with no reported serious or severe AEs.

In parallel, promising findings for PRA023 were revealed from a phase 2 placebo-controlled multicenter, double-blind study assessing the efficacy of this drug in moderate-to-severe active UC patients (ARTEMIS-UC) [84]. The study employed a 1:1 randomization of patients to receive either iv PRA023 (1000 mg on day 1, 500 mg at weeks 2, 6, and 10) or placebo. At week 12, a significantly greater proportion of patients treated with PRA023 achieved clinical remission and endoscopic improvement compared to the placebo group (26.5% PRA023 vs. 1.5% placebo, *p* < 0.0001 and 36.8% PRA023 vs. 6.0% placebo, *p* < 0.0001, respectively). No serious AEs were reported. To validate these promising results, a phase 3 study will be conducted.

### 3.7. PDE Inhibitor

Cyclic nucleotide phosphodiesterases (PDEs) are a large family of enzymes that catalyze the hydrolysis of cAMP and/or cGMP. PDE4, in particular, is expressed in dendritic cells, macrophages, monocytes, and T cells, and is considered an important player in the inflammatory response. PDE4 indirectly enhances the activation of NF-κB, promotes the production of proinflammatory mediators such as tumor necrosis factor-a (TNF-a), IL-23, IL-17, and interferon-gamma (IFN-γ), and decreases the expression of anti-inflammatory cytokines such as IL-10. Thus, PD4 inhibitors may represent a therapeutic target in IBD by acting at multiple levels [54].

#### Apremilast

Apremilast, an oral small-molecule PDE4 inhibitor, acts intracellularly through the inhibition of TNF-a and matrix metalloproteinase 3 (MMP-3).

A multicenter, randomized, double-blind, phase 2 trial was conducted on 170 adult patients with active UC for 3 months or more who were naïve, could not tolerate or failed biologic therapy, or had contraindications to conventional therapies [85]. Apremilast at doses of 30 mg and 40 mg twice daily was compared to placebo for 12 weeks, followed by an additional 40 weeks of apremilast treatment. While the primary endpoint of achieving clinical remission at week 12 was not met, a higher proportion of patients treated with apremilast (30 mg or 40 mg) showed improvements in clinical features and markers of inflammation compared to placebo. Specifically, 31.6% of patients in the 30 mg group and 21.8% in the 40 mg group achieved clinical remission, compared to 12.1% in the placebo group. These improvements were sustained up to week 52 for patients who continued apremilast treatment. The most commonly reported AEs were headache and nausea.

### 3.8. TLR9 Agonist

Toll-like receptors (TLRs), act like sentinels to protect the host from external microbial invasion [86]. TLR9, which is a member of the TLR family, binds DNA containing unmethylated immunostimulatory dinucleotide CpGs of bacterial or viral origin. TLR9’s role in the pathogenesis of colitis has been studied in murine experimental models [87,88]. Interestingly, TLR9-deficient mice developed more severe colitis compared to wild-type controls. The severity of colitis was significantly reduced following treatment with TLR9 agonist. The TLR9 pathway protected the epithelial barrier, induced T regulatory cells (Tregs), and increased anti-inflammatory cytokine IL-10 expression. Thus, TLR9 agonists may have a crucial role in IBD treatment [55].

#### Cobitolimod

Cobitolimod (DIMS0150) is a synthetic oligodeoxynucleotide (ODN) that contains an unmethylated CpG motif and activates TLR9 in specific cells, including intestinal T and B lymphocytes and antigen-presenting cells (APCs), leading to the production of anti-inflammatory cytokines such as interleukin-10 (IL-10) and type I interferons.

This topically administered drug was recently tested in a randomized, phase 2b study in patients with moderate-to-severe, left-sided UC (CONDUCT study) [89]. The trial included 213 UC patients with a Mayo Score of 6–12 (endoscopy subscore of 2 or higher) who failed to respond to conventional or biological therapies. The patients were randomized into five groups: cobitolimod 31 mg, 125 mg, or 250 mg at weeks 0 and 3, cobitolimod 125 mg at weeks 0, 1, 2, and 3, or placebo. At week 6, the group receiving two administrations of cobitolimod 250 mg showed a significantly higher proportion of patients achieving clinical remission (Mayo Subscores for rectal bleeding of 0, for stool frequency of 0 or 1, and for endoscopy of 0 or 1) compared to the placebo group (21% versus 7%). No significant differences were observed in the other treatment groups. The authors concluded that two topical administrations of cobitolimod 250 mg were effective and safe in inducing clinical remission in patients with active left-sided UC.

Cobitolimod holds promise as a novel and unique therapeutic approach in patients with UC, with a currently ongoing large-scale phase 3 trial (NCT04985968).

### 3.9. Selective Upregulation of miR-124 Expression

In the past few years, miRNAs have been demonstrated to play crucial roles as modulators of gene expression in different biological processes such as the development of the immune system and the regulation of both innate and adaptive immune responses. Specifically, some studies showed that miRNAs could regulate intestinal epithelial tight junction permeability, intestinal IL-12/IL-23p40 expression, Th-17 cell differentiation, and inflammatory cell trafficking [90,91,92,93,94]. Moreover, a recent study by Koukos et al. demonstrated that miR-124, which is a modulator of monocyte and macrophage activation, could be involved in the pathogenesis of UC. In fact, reduced levels of miR-124 have been identified in the colonic mucosa of pediatric patients with UC, which could promote inflammation through increasing expression and activity of STAT3 [56].

#### Obefazimod

ABX464 (Obefazimod) is a quinoline that selectively upregulates miR-124 in immune cells and reverses the expression of several inflammatory cytokines triggered during inflammation. This small molecule has therefore been proposed for the treatment of moderate-to-severe UC patients and was recently tested in phase 2a and phase 2b studies [57,95].

The phase 2a study included 32 adult patients with moderate-to-severe UC and comprised an 8-week, placebo-controlled, double-blind induction phase followed by an open-label long-term extension phase. During the induction phase, patients received 50 mg ABX464 orally or placebo once daily for 8 weeks. In the long-term extension, all patients received ABX464 50 mg once daily. At week 8, the ABX464 group showed higher rates of clinical remission (70%) and clinical response (35%) compared to the placebo group (33% and 11%, respectively). After 12 months of ABX464 treatment, 75% of patients with an assessable endoscopy achieved clinical remission, 33.3% maintained sustained remission, and 66.7% acquired clinical remission during the long-term extension phase. AEs were reported in 78% of patients in the ABX464 group compared to 55% in the placebo group, with abdominal pain and headache being the most common.

In the double-blind, randomized, placebo-controlled phase 2b study, patients were administered ABX464 (25 mg, 50 mg, or 100 mg) or placebo once daily. Clinical results at week 8 demonstrated that all doses of ABX464 (25, 50, or 100 mg) once daily led to clinical remission compared to placebo. Alongside its efficacy, ABX464 exhibited a favorable tolerability and safety profile.

ABX464 appears to have the ability to reverse the expression of various inflammatory cytokines associated with inflammation and exhibits promising, well-tolerated, safe, and long-lasting effects on moderate-to-severe active UC compared to placebo. A phase 3 clinical program is ongoing.

### 3.10. Anti-IP-10

Interferon-γ-inducible protein-10 (IP-10), also known as CXCL10, is a chemokine that plays an important role in the activation of proinflammatory pathways through the interaction with chemokine receptor 3 (CXCR3). IP-10 is also implicated in epithelial cell proliferation and migration, independently of CXCR3 [58,96,97,98].

#### BMS-936557

A recent 8-week randomized, double-blind, multicenter, phase 2 study was performed to evaluate a fully human monoclonal antibody that targets interferon-g-inducible protein 10 (IP-10) for the treatment of moderate-to-severe UC [99]. A total of 109 patients were selected to receive placebo or BMS-936557 (10 mg/kg) iv at weeks 0, 2, 4, and 6 (BMS *n* = 55, placebo *n* = 54). BMS-936557 was generally safe and well tolerated. Primary and secondary endpoints were not reached (respectively, clinical response at day 57 and clinical and endoscopic remission); however, post hoc analyses showed that higher BMS-936557 steady-state trough concentration was associated with enhanced clinical response. Further dose–response studies are required to evaluate the efficacy of the antibody.

## 4. Conclusions

Several novel small molecules and biological drugs have recently undergone phase 2 and 3 clinical trials to modulate inflammatory and molecular pathways in patients with active moderate-to-severe IBD. Encouraging results have been observed with some compounds, suggesting their potential use in routine therapy in the future. Notably, JAK inhibitors and S1PR modulators have shown promising results in clinical trials. To mitigate systemic toxicity associated with JAK inhibitors, gut-selective options like TD-1473 have been explored, despite primary outcomes not being met. S1PR modulators have a unique mechanism of action involving lymphocyte trapping in lymphoid organs, making them potentially effective for IBD, particularly UC. Selective IL-23p19 monoclonal antibodies, including mirikizumab, have demonstrated excellent efficacy and safety profiles also in biologic-experienced patients. Furthermore, orally administered α4 integrin and α4β7 antagonists offer simplicity of administration compared to iv/sc biological agents and do not pose the challenges of immunogenicity and antibody development associated with biological therapies. In addition, cobitolimod holds potential as a topical therapy for left-sided UC. Finally, PRA023, an anti-TL1A monoclonal antibody with an anti-fibrotic mechanism of action, recently showed interesting results in CD patients.

However, challenges persist as not all studies have achieved their primary endpoints. Well-designed, large, controlled phase 3 clinical trials are still necessary for most drugs.

Finally, the development of specific therapeutic models through a multitarget approach will enable personalized therapy for patients with IBD in the future.

## Figures and Tables

**Figure 1 biomedicines-11-02249-f001:**
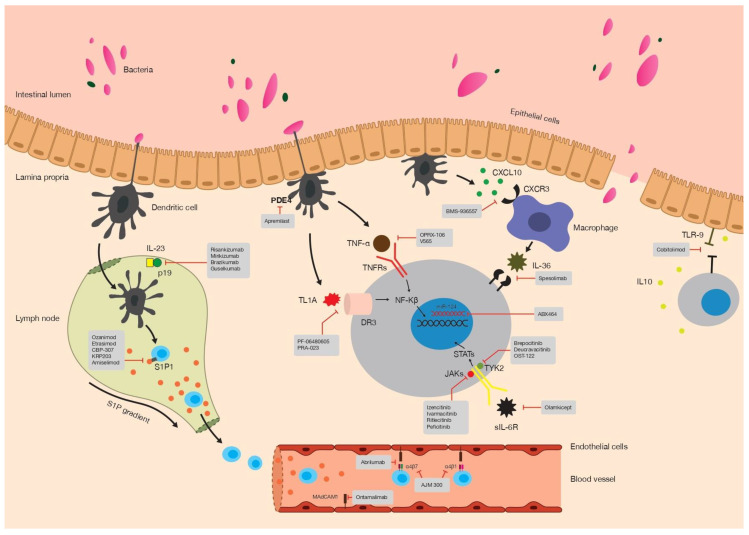
Mode of action of promising drugs in phase 2/3 trials for UC and CD patients.

**Table 1 biomedicines-11-02249-t001:** Efficacy and safety data of promising phase 2/3 novel therapies in CD. Clinical trials currently ongoing or under development are also mentioned.

Class	Drug [Ref]	Route of Administration	Study Type	Study Population	Study Design	Primary Endpoint	Results of Primary Endpoint	AEs, SAEs, and Deaths	Trials Currently under Development
IL-23 inhibitors	Brakizumab (MEDI2070) [14]	IV/SC	Phase 2a	119 moderate-to-severe active CD patients with previous anti-TNF failure	Double-blind, randomized, placebo-controlled 12-week induction phase (1:1 randomization to placebo or MEDI2070 IV 700 mg) and a 100-week open-label phase (MEDI2070 SC 210 mg/4 weeks)	Clinical response at week 8, specified as either a 100-point decrease in CDAI score from baseline or clinical remission characterized as CDAI score < 150	Primary endpoint occurred in 49.2% of patients in the MEDI2070 group vs. 26.7% in the placebo group (*p* = 0.010)	Most common AEs: headache and nasopharyngitis; SAEs: 8.5% in the MEDI2070 group vs. 8.3% in the placebo group; Deaths: no	A 52-week, multicenter, randomized, double-blind, placebo and active-controlled, operationally seamless phase 2b/3 trial (INTREPID) on 89 patients with severely active CD
	Guselkumab [15,16,17]	IV/SC	Phase 2	309 moderate-to-severe CD patients	Multicenter, double-blind, randomized, placebo-controlled treat-through design over 48 weeks Patients were randomized 1:1:1:1:1 to receive either guselkumab at dosages of 200 mg, 600 mg, 1200 mg or placebo at weeks 0, 4, and 8, or ustekinumab (reference arm) 6 mg/kg IV at week 0 and 90 mg SC at week 8.	Clinical response at week 12 defined by a decrease in CDAI score	Results regarding primary endpoint were significantly higher in patients treated with guselkumab 200 mg and 400 mg vs. placebo (61.4% and 60.7% vs. 27.6%, respectively, both *p* < 0.001)	Most common AEs: headache and nasopharyngitis; Reported AEs and SAEs were not greater compared with placebo; Deaths: no	Phase 3 open-label trials Phase 2 trial to assess the efficacy and safety of guselkumab and golimumab combination therapy
	Mirikizumab [18]	IV/SC	Phase 2	191 moderate-to-severe CD patients 2/3 of them were previously exposed to anti-TNF and almost 50% experienced anti-TNF prior failure	Multicenter, parallel-arm, double-blind, placebo-controlled trial Patients were randomized (2:1:1:2) to be administered placebo or 200 mg, 600, or 1000 mg of mirikizumab IV at weeks 0, 4, and 8. Patients who received mirikizumab and achieved ≥ 1 point improvement in SES- CD at week 12 were re-randomized 1:1 to mirikizumab IV every 4 weeks or mirikizumab 300 mg SC every 4 weeks until week 52	Endoscopic response at week 12, defined as a 50% reduction from baseline in SES-CD	Primary endpoint was reached in 25.8% of the 200 mg group (*p* = 0.079); 37.5% of the 600 mg group (*p* = 0.003); 43.8% of the 1000 mg group (*p* < 0.001)	Most common AEs: headache, arthralgia, nasopharyngitis, increased weight, and nausea; SAEs: 0 in the 200 mg group; 3 in the 600 mg group and 2 in the 1000 mg group during the induction phase; 0 in the IV group and 3 in the 300 mg SC group in the maintenance period; Deaths: no	Phase 3 trials in adult and pediatric patients are ongoing
IL-36 inhibitor	Spesolimab [19,20]	IV							Phase 2 clinical trials in CD patients with fistulizing disease
IL-6 trans-signaling inhibitor	Olamkicept [21]	IV	Phase 2a	16 IBD patients (7 with moderately to severely active CD)	12-week, open-label study Patients were given olamkicept 600 mg iv every 2 weeks	Clinical remission defined as a CDAI score < 150	Primary endpoint was achieved in 14.2% of patients with CD	Most common AEs: seasonal upper respiratory tract infections, recurrence of herpes labialis, eczema, erythema; SAEs: 31% of patients (5/16); Deaths: no	A placebo-controlled, larger clinical study is underway (NCT03235752) to further investigate whether this class of drug does not cause any suppression in humans
Anti-TNF	V565 [22]	Oral	Phase 2	125 moderate-to-severe CD patients	6 weeks double-blind, placebo-controlled, parallel-group Patients were randomized 2:1 to receive V565 or placebo 3 times a day	Clinical response at day 42, defined as a 70-point reduction in CDAI score and a reduction of inflammatory markers from baseline (>40% decrease from baseline in protein C-reactive or fecal calprotectin)	Primary endpoint was not achieved	Most common AEs: N.A.; SAEs: 3.66% in the V565 group vs. 4.65% in the placebo group; Deaths: no	
Anti-adhesion	AJM-300 [23]	Oral	Phase 3b	539 moderate-to-severe CD patients with prior anti-TNF failure	6-week open-label induction phase (AJM-300 400 mg at weeks 0, 2, and 4) followed by a 20-week double-blind maintenance phase (AJM-300 400 mg every 2 or 4 weeks)	Clinical response at week 6 (CDAI decrease >100 from baseline)	62% of patients reached primary endpoint	N.A.	
	Ontamalimab (OPERA I) [24]	SC	Phase 2	265 CD patients with history of failure or intolerance to anti-TNF and/or immunosuppressive agents, high-sensitivity C reactive protein > 3.0 mg/L, and ulcers on colonoscopy	Multicenter randomized double-blind, placebo-controlled, parallel-group phase 2 trial Patients were randomized (1:1:1:1) to receive ontamalimab at dosages of 22.5 mg, 75 mg, or 225 mg or placebo at weeks 0, 4, and 8 and were followed through 12 weeks	Clinical response at week 8 or 12 defined as a 70-point decrease in CDAI score	Primary endpoint was not achieved	Most common AEs were related to the underlying disease; SEAs: 16.7% (22.5 mg group), 13.8% (75 mg group), 16.2% (225 mg group), 7.9% (placebo group); Deaths: no	
	Ontamalimab (OPERA II) [25]	SC	Phase 2	268 CD patients who had a clinical response in the OPERA I study or in the TOSCA study	72-week, multicenter, open-label phase 2 extension study, assessing the long-term safety and efficacy of ontamalimab Patients received ontamalimab 75 mg SC every 4 weeks until week 72 with a 24-week follow-up period	The primary endpoint were safety and tolerability outcomes	Ontamalimab was well tolerated. 149/268 patients completed the study. The most common AE or SAE leading to interruption of the treatment was CD flare.	Most common AEs: CD flare; SAEs: 10 patients; Deaths: two, unrelated to the drug	
S1P modulators	Ozanimod [26]	Oral	Phase 2	60 moderate-to-severe CD patients	Uncontrolled multicenter trial comprising a 12-week induction phase where patients underwent a 7-day dose escalation period followed by a 100-week extension phase where patients received ozanimod 1 mg daily	Endoscopic response at week 12 defined as a change in SES-CD from baseline to week 12	23·2% of patients reached primary endpoint	Most common AEs: CD flare; SAEs: CD exacerbation (9%) and abdominal abscess (3%); Deaths: no	YELLOWSTONE phase 3 program comprising induction and maintenance trials and an open-label extension study to assess the safety and efficacy of ozanimod in patients with moderately to severely active CD
	Amiselimod [27]	Oral	Phase 2a	180 moderate-to-severe CD patients	Multicenter, double-blind, placebo-controlled Patients were randomized 1:1 to: amiselimod 0.4 mg daily vs. placebo over 14 weeks	Clinical response at week 12, defined as a 100-point decrease from baseline in the CDAI score	Primary endpoint was not achieved	Mos common AEs: headache (13%), nasopharyngitis, and arthralgia (both 6.5%); SAEs: 7 participants in the amiselimod group, 4 discontinued treatment; Deaths: no	
JAK inhibitors	Izencitinib [28]	Oral	Phase 2	304 moderate-to-severe CD patients with corticosteroid-dependence or prior failure to conventional therapies	Multicenter, double-blind, placebo-controlled Patients were randomized 2:3:3 to: placebo or izencitinib 80 mg or 200 mg once daily for 12 weeks	Change in CDAI score from baseline to week 12	Primary endpoint was not achieved	N.A.	
	Ivarmacitinib [29]	Oral	Phase 2	112 moderate-to-severe-CD patients	Multicenter, double-blind, placebo-controlled (12 + 12)-week design	Clinical remission at week 12 defined as a CDAI score < 150	Pending		
Anti-TL1AR	PRA-023 [30]	IV	Phase 2a	55 moderate-to-severe active CD patients with high-rate of prior biologic exposure (70.9%) and a mean disease duration of 10.3 years	Open-label PRA-023 1000 mg on day 1, 500 mg at weeks 2, 6 and 10	Endoscopic response at week 12, defined as a reduction in SES-CD score of >50%	Primary endpoint was achieved in 26% PRA-023 vs. 12% placebo (*p* = 0.002)	N.A.	

Ref = reference; IV = intravenous; SC = subcutaneous; SES-CD = Simple endoscopic score for CD; CDAI = CD activity index; SAEs = severe adverse events; N.A. = non-reported.

**Table 2 biomedicines-11-02249-t002:** Efficacy and safety data of promising phase 2/3 novel therapies in UC. Clinical trials currently ongoing or under development are also mentioned.

Class	Drug [Ref]	Route of Administration	Study Type	Study Population	Study Design	Primary Endpoint	Results of Primary Endpoint	AEs, SAEs, and Deaths	Trials Currently under Development
IL-23 inhibitors	Guselkumab (QUASAR Induction study) [15]	IV/SC	Phase 2b	313 moderate-to-severe UC	Multicenter, randomized, double-blind, placebo-controlled Patients were randomly assigned to guselkumab at 200 mg every 4 weeks, or guselkumab at 400 mg every 4 weeks	12-week clinical remission rates	Primary endpoint was reached in 61.4% for the lower dose of guselkumab and 60.7% for the higher dose of guselkumab, both significantly higher than placebo (*p* < 0.001)	AEs: 1% in the guselkumab arms vs. 5.7% in the placebo arm; SAEs: N.A.; Deaths: no	Phase 3 ongoing
	Guselkumab + Golimumab (VEGA trial) [31]	IV/SC	Phase 2a	214 moderate-to-severe UC patients naïve to anti-TNF	Patients were randomly assigned to guselkumab iv 200 mg at weeks 0, 4, and 8 (*n* = 71); golimumab 200 mg sc at week 0 and then 100 mg at weeks 2, 6, and 10 (*n* = 72); or a combination of these treatment regimens (*n* = 71)	12-week clinical remission rates	Primary endpoint was reached in 83% of patients in dual therapy vs. 74% in the guselkumab group and 61% in the golimumab group	AEs: N.A.; SAEs: 4.2% in the golimumab arm, 1.4% in the guselkumab arm, and 2.8% in the combination arm; Deaths: no	
	Mirikizumab [32]	IV/SC	Phase 3 induction	1281 moderate-to-severe UC patients	Multicenter, randomized, double-blind, placebo-controlled Patients were randomly assigned to mirikizumab 300 mg or placebo, iv, every 4 weeks for 12 weeks	12-week clinical remission rates	Primary endpoint was reached in 24.2% vs. 13.3%in the placebo group, *p* < 0.001		
	Mirikizumab [32]	IV/SC	Phase 3 maintenance	544 moderate-to-severe UC	Patients with a response in induction therapy were randomized to receive mirikizumab 200 mg or placebo, sc, every 4 weeks for 40 weeks	52-week clinical remission rates	Primary endpoint was reached in 49.9% vs. 25.1% in the placebo group, *p* < 0.001	AEs: nasopharyngitis and arthralgia; SAEs: 15 had an opportunistic infection (including 6 with herpes zoster infection) and 8 had cancer (including 3 with colorectal cancer)	
IL-36 inhibitor	Spesolimab [33,34]	IV	Phase 2/2a	moderate-to-severe UC patients			Efficacy endpoints were not met	N.A.	
Selective inhibitors of IL-6 trans-signaling	Olamkicept [21]	IV	Phase 2a	16 IBD patients (9 with moderately to severely active UC)	12 weeks, open-label study Patients were given olamkicept 600 mg IV every 2 weeks	12-week clinical response rates	Primary endpoint was reached in 22.2% of patients with UC	Most common AEs: seasonal upper respiratory tract infections, recurrence of herpes labialis, eczema, erythema; SAEs: 31% of patients (5/16); Deaths: no	A placebo-controlled, larger clinical study is underway (NCT03235752) to further investigate whether this class of drug does not cause any suppression in humans
	Olamkicept [35]	IV	Phase 2	90 moderate-to-severe UC patients	Randomized, double-blind, placebo-controlled Patients were randomly assigned to olamkicept 300 mg, 600 mg, or placebo IV every 2 weeks	12-week clinical response rates	Primary endpoint was reached in 58.6% of patients in the 600 mg group (*p* = 0.03) and 43.3% of patients in the 300 mg group (*p* = 0.52) vs. 34.5% in the placebo group	Most common AEs: bilirubin presence in the urine, hyperuricemia, and increased aspartate aminotransferase levels; SAEs: N.A.; Deaths: no	
Anti-TNF	OPRX-106 [36]	Oral	Phase 2a	25 moderate-to-severe UC patients	Open-label Patients were randomly assigned to either 2 or 8 mg of OPRX-106 once daily for a duration of 8 weeks	8-week clinical remission rates	Primary endpoint was reached in 67% of patients	Most common AEs: headache, nausea, fatigue, anemia; SAEs: N.A; Deaths: no	
Anti-adhesion	Abrilumab [37]	SC	Phase 2b	354 moderate-to-severe UC patients	Multicenter, randomized, double-blind, placebo-controlled Patients were randomly assigned to abrilumab (7, 21, or 70 mg) on day 1, weeks 2 and 4, and every 4 weeks; abrilumab 210 mg on day 1; or placebo	8-week clinical remission rates	Primary endpoint was reached in 4% placebo, 13% abrilumab 70 mg, and 12% abrilumab 210 mg	Most common AEs: non-serious infections, headache, and arthralgia; SAEs: 12.1% in the placebo group, 5% in the 7 mg group, 7.5% in the 21 mg group, 5.1% in the 70 mg group, and 8.9% in the 210 mg group Deaths: no	
	Abrilumab [38]	SC		45 moderate-to-severe UC patients	randomized, double-blind, placebo-controlled Patients were randomly assigned abrilumab 21 mg, 70 mg, or 210 mg, for 12 weeks followed by a 36-week open-label period (abrilumab 210 mg every 12 weeks)	12-week clinical remission rates	Primary endpoint was reached in 12.9% of patients in the abrilumab groups vs. 0% in the placebo group	Most common AEs: headache, malaise, and asthma; SAEs: 10% in the 21 mg group and 11.1% in the 210 mg group; Deaths: no	
	AJM-300 [39]	Oral	Phase 2a	102 moderate-to-severe UC patients	Multicenter, randomized, double-blind, placebo-controlled Patients were randomly assigned to AJM-300 960 mg or placebo three times daily for 8 weeks	8-week clinical remission rates	Primary endpoint was reached in 62.7% of the treatment group vs. 23.5% of the placebo group	Most common AEs: nasopharyngitis and worsening of UC; SAEs: N.A.; Deaths: no	
	AJM-300 [40]	Oral	Phase 3	203 moderate-to-severe UC patients	Multicenter, randomized, double-blind, placebo-controlled Patients were randomly assigned to AJM-300 960 mg or placebo three times daily for 8 weeks, followed by a 24-week open-label re-treatment phase	8-week clinical response rates	Primary endpoint was reached in 45% of patients in the AJM-300 group vs. 21% of patients in the placebo group (*p* = 0.00028)	Most common AEs: nasopharyngitis and worsening of UC; SAEs: one in the AJM-300 group; Deaths: no	
	Ontamalimab [41]	SC	Phase 3	587 moderate-to-severe UC patients with prior failure to conventional therapy	Multicenter, randomized, double-blind, placebo-controlled Patients were randomly assigned to 7.5 mg, 22.5 mg, and 75 mg or 225 mg Ontamalimab or placebo administered at baseline and then every 4 weeks or placebo	12-week clinical remission rates	7.5 mg (*p* = 0.0425), 22.5 mg (*p* = 0.0099), and 75 mg(*p* = 0.0119), 225 mg (*p* = 0.1803) Ontamalimab vs. placebo	Most common AEs: headache and nasopharyngitis SAEs: 5.5% in the placebo group; 15.5% in the 7.5 mg group; 1.4% in the 22.5 mg group; 4.1% in the 75 mg group and 4.3% in the 225 mg group Deaths: no	
S1P modulators	Ozanimod [42]	Oral	Phase 3	645 moderate-to-severe UC patients	Multicenter, randomized, double-blind, placebo-controlled Patients were randomly assigned to ozanimod 1 mg per day or placebo	12-week clinical remission rates	Primary endpoint was reached in 18.4% of patients in the etrasimod group vs. 6% in the placebo group (*p* = 0.001)	Most common AEs: N.A SAEs: Less than 2% in each group Deaths: one occurred in a patient with a history of ischemic cardiomyopathy and prolonged tobacco use	
	Etrasimod (ELEVATE 12) [43]	Oral	Phase 3	354 moderate-to-severe CD patients	Multicenter, randomized, double-blind, placebo-controlled Patients were randomly assigned to etrasimod 2 mg or placebo for 12 weeks	12-week clinical remission rates	Primary endpoint was reached in 25% of patients in the etrasimod group vs. 15% in the placebo group (*p* = 0.026)	Most common AEs: anemia, headache, and worsening of UC; SAEs: 3% in the etrasimod group vs. 2% in the placebo group Deaths: no	
	(ELEVATE 52) [43]	Oral	Phase 3	433 moderate-to-severe UC patients	Multicenter, randomized, double-blind, placebo-controlled Patients were randomly assigned to etrasimod 2 mg or placebo for 12 weeks followed by a 40-week maintenance period with a treat-through design	12- and 52-week clinical remission rates	Week 12: primary endpoint was reached in 27% of patients in the etrasimod group vs. 7% of patients in the placebo group (*p* < 0.0001); Week 52: primary endpoint was reached in 32% of patients in the etrasimod group vs. 7% of patients in the placebo group (*p* < 0.0001)	Most common AEs: anemia, headache, and worsening of UC; SAEs: 7% in the etrasimod group vs. 6% in the placebo group Deaths: no	
	CBP-307 [44]	Oral	Phase 2	145 moderate-to-severe UC patients	Multicenter, randomized, double-blind, placebo-controlled Patients were randomly assigned to CBP-307 0.1 mg, CBP-307 0.2 mg, or placebo	12-week clinical remission rates	Primary endpoint was not reached	Most common AEs: N.A. SAEs: 3.8% in the CBP-307 0.2 mg group vs. 5.8% in the placebo group Deaths: N.A.	
	KRP203 [45]	Oral	Phase 2	72 patients with moderately active 5-aminosalicylate-refractory UC	Multicenter, randomized, double-blind, placebo-controlled Patients were randomly assigned to 1.2 mg KRP203 or placebo daily for 8 weeks	8-week clinical remission rates	Primary endpoint was reached in 14% of KRP203 group vs. 0% of placebo group	Most common AEs: headache and diarrhea; SAEs: 2 patients in the KRP203 group vs. 5 patients in the placebo group; Deaths: no	
JAK inhibitors	Izencitinib [46]	Oral	Phase 2b	239 moderate-to-severe UC patients	Multicenter, randomized, double-blind, placebo-controlled	8-week clinical remission rates	Efficacy endpoints were not met	AEs and SAEs: N.A.; Deaths: no	
	Ivarmacitinib [47]	Oral	Phase 2	146 moderate-to-severe UC patients	Multicenter, randomized, double-blind, placebo-controlled Patients were randomly assigned to the following treatment groups: ivarmacitinib 8 mg once daily, 4 mg twice daily, or 4 mg once daily, or placebo for 8 weeks	8-week clinical response rates	Primary endpoint was significantly higher in the 8 mg once daily group (46.3%; P = 0.066), 4 mg twice daily group (46.3%; P = 0.059), and 4 mg once daily group (43.9%; P = 0.095) vs. placebo (26.8%)	AEs and SAEs: N.A.; Deaths: no	A phase 3 study is underway
	Peficitinib [48]	Oral	Phase 2b	219 moderate-to-severe UC patients	Dose-ranging placebo-controlled trial Patients were randomly assigned to the following treatment groups: peficitinib at 25 mg, 75 mg, or 150 mg once daily, or peficitinib 75 mg twice daily versus placebo once daily	8-week clinical dose–response rates	Primary endpoint was not reached	Most common AEs: worsening of UC; SAEs: 3.4% in the combined peficitinib group vs. 4.7% in the placebo group; Deaths: no	
	Ritlecitinib [49]	Oral	Phase 2b	317 moderate-to-severe UC patients	Randomized, double-blind, placebo-controlled trial conducted over 34 weeks Patients were randomly assigned to the following treatment groups: ritlecitinib (20 mg, *n* = 51; 70 mg, *n* = 49; 200 mg, *n* = 50), brepocitinib (10 mg, *n* = 48; 30 mg, *n* = 47; 60 mg, *n* = 47; or placebo for 8 weeks	8-week clinical remission rates	Clinical remission was significantly higher in in the ritlecitinib 70 and 200 mg, *p* < 0.001 and *p* < 0.001, respectively	Most common AEs: anemia, headache, nasopharyngitis, abdominal pain, pyrexia, and arthralgia; SAEs: N.A.; Deaths: two, considered unrelated to study drug	
	Brepocitinib [49]	Oral	Phase 2b	317 moderate-to-severe UC patients	Randomized, double-blind, placebo-controlled trial conducted over 34 weeks Patients were randomly assigned to the following treatment groups: ritlecitinib (20 mg, n = 51; 70 mg, *n* = 49; 200 mg, *n* = 50), brepocitinib (10 mg, *n* = 48; 30 mg, *n* = 47; 60 mg, *n* = 47; or placebo for 8 weeks	8-week clinical remission rates	Clinical remission was significantly higher brepocitinib 30 and 60 mg groups, *p* = 0.001 and *p* < 0.001, respectively	Most common AEs: anemia, headache, nasopharyngitis, abdominal pain, pyrexia, and arthralgia; SAEs: N.A.; Deaths: two, considered unrelated to study drug	
	Deucravacitinib [50]	Oral	Phase 2	131 moderate-to-severe UC patients	Randomized, double-blind, placebo-controlled Patients were randomly administered deucravacitinib 6 mg or placebo twice daily	12-week clinical remission rates	Primary endpoint was not reached	Most common AEs: N.A.; SAEs: 9.2% of patients in the deucravacitinib arm; Deaths: no	A second phase 2 trial will evaluate a higher dose of deucravacitinib in patients with UC
	OST-122 [51]	Oral	Pase1b/2a	32 moderate-to-severe UC patients	Randomized, double-blind, placebo-controlled trial Patients were randomly given OST-122 or placebo once daily over 28 days	Safety and tolerability of OST-122	Pending		
Anti-TL1AR	PF-06480605 [52]	IV	Phase 2a	50 moderate-to-severe UC patients	Multicenter, single-arm, open-label study All patients received 500 mg iv PF-06480605 every 2 weeks, 7 doses total, with a 3-month follow-up period	14-week endoscopic improvement	Primary endpoint was reached in 38.2% of patients, *p* = 0.001	Most common AEs: UC exacerbation and arthralgia; SAEs: N.A.; Deaths: no	
	PRA-023 [53]	IV	Phase 2	135 moderate-to-severe UC patients	Randomized, double-blind, placebo-controlled Patients were randomly administered iv PRA023 (1000 mg on day 1, 500 mg at weeks 2, 6, and 10) or placebo	12-week clinical remission rates	Primary endpoint was reached in 26.5% of PRA023 patients vs. 1.5% of placebo patients, *p* < 0.0001	SAEs: N.A. Deaths: no	Phase 3 trials will be conducted
PDE4 inhibitor	Apremilast [54]	Oral	Phase 2	170 moderate-to-severe UC patients	Randomized, double-blind, placebo-controlled For the 12-week phase, patients were randomly administered apremilast 30 mg (*n* = 57), apremilast 40 mg (*n* = 55), or placebo (*n* = 58) twice daily; patients were then randomly assigned to groups that received apremilast, 30 or 40 mg twice daily, for an additional 40 weeks	12-week clinical remission rates	Primary endpoint was reached in 31.6% of patients in the 30 mg apremilast group vs. 12.1% of patients in the placebo group, *p* = 0.01	Most common AEs: headache and nausea; SAEs: 0% in the 30 mg twice daily group, 1.8% in the 40 mg twice daily group, and 3.4% in the placebo group; Deaths: no	
TLR9 agonist	Cobitolimod [55]	Topical	Phase 2b	213 moderate-to-severe left-sided UC patients	Randomized, double-blind, five-arm, placebo-controlled, dose-ranging Patients were randomly administered rectal enemas of cobitolimod at 31 mg, 125 mg, or 250 mg at weeks 0 and 3 (2 × 31 mg, 2 × 125 mg, and 2 × 250 mg groups), cobitolimod at 125 mg at weeks 0, 1, 2, and 3 (4 × 125 mg group), or placebo	6-week clinical remission rates	A greater proportion of patients were in clinical remission at week 6 in the cobitolimod 2 × 250 mg group than in the placebo group (21% vs. 7%; *p* = 0.025)	Most common AEs: worsening of UC; SAEs: 5% in the placebo group, 5% in the cobitolimod 2 × 31 mg group, 5% in the 4 × 125 mg and 10% in the 2 × 250 mg group Deaths: one patient in the placebo group died from total organ failure after receiving a colectomy for a serious adverse event of disease worsening.	Phase 3 trial (NCT04985968) is ongoing
Selective upregulation of miR-124 expression	Obefazimod (ABX464) [56]	Oral	Phase 2a	32 moderate-to-severe UC patients	Randomized, double-blind, placebo-controlled Patients were randomly administered ABX464 50 mg or placebo once daily	8-week clinical remission rates	70% in the ABX464 vs. 33% in the placebo group	Most common AEs: abdominal pain and headache; SAEs: N.A.; Deaths: no	
	Obefazimod (ABX464) [57]	Oral	Phase 2b	254 moderate-to-severe UC patients	Randomized, double-blind, multicenter, placebo-controlled Patients were randomly administered ABX464 (25 mg, 50 mg, 100 mg) or placebo once daily	8-week clinical remission rates	all doses of ABX464 (25, 50, or 100 mg) once daily led to clinical remission compared to placebo (*p* = 0.0039 for ABX464 100 mg vs. placebo; *p* = 0.0003 for ABX464 50 mg vs. placebo; and *p* = 0.0010 for ABX464 25 mg vs. placebo)	Most common AEs: headache; SAEs: one in each of the ABX464 100 mg and 50 mg groups; Deaths: no	Phase 3 trial is ongoing
Anti-IP-10	BMS-936557 [58]	IV	Phase 2	109 moderate-to-severe UC patients	8-week randomized, double-blind, multicenter, placebo-controlled Patients were randomly administered BMS-936557 (10 mg/kg) iv or placebo at weeks 0, 2, 4, and 6	8-week clinical response	Primary endpoint was not achieved	N.A.	

Ref = reference; IV = intravenous; SC = subcutaneous; SAEs = severe adverse events; N.A. = non-reported.

## Data Availability

Not applicable.

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
