# Peer review of "Inflammatory Bowel Disease: Emerging Therapies and Future Treatment Strategies"

_biomedicines, 2023, doi:10.3390/biomedicines11082249_

Round 1
Reviewer 1 Report
Bretto E and collaborators review the results of phase 2 and 3 clinical trials in the new treatments for Crohn´s disease and ulcerative colitis in terms of clinical response/remission, endoscopic score, and adverse effects. The idea of the review is interesting since no reviews in this line are in the literature. The authors conducted a very complete review accompanied by a huge number of recent references very interesting and useful for the reader. The figure is very useful for the reader. The table should be improved.
Major points
Apart from the small size of the words, the content of Table 1 is not understood, why do the authors add only some of the clinical trials instead of all the clinical trials that they describe throughout the text? They should describe the title in each column better. They should include in the table all the clinical trials that they include in the manuscript.
-Parameters to define clinical remission or response (CDAI, Mayo score, etc.) are required in all the studies. Authors include them only in some studies.
-Parameters/symptoms on which CDAI, Mayo score, Baron Score, CDAI-100, Robarst histology index, and CD-SES are based should add in the text or maybe in a table. They should be useful for the readers.
-Countries and companies involved in the clinical trials described would be very interesting for the readers. Even in the table. Additionally, if the same group/company has carried out a clinical trial before this information should be included in the text.
-Authors should revise abbreviations throughout the manuscript, e.g. lines 103, 521, 574, 631, 72 etc. If they define some words as abbreviations such as adverse events, intravenous, Crohn´s disease, etc. they should use the abbreviations in the rest of the manuscript.
Minor points
In general:
-Keep the same pattern for “iv” and “sc” throughout the text. I recommend capital letters for both. Additionally, the letters in the phase of clinical trials, e.g. line 554 “phase 2b” instead of “phase 2B”, line 576.
-Authors should be consistent with the abbreviations “AEs” and “SAEs”, “Mayo score” or “Mayo Clinic score”
-Authors should include a space between numbers as doses throughout the text.
In details,
-A brief description of Risankzumab should be added in line 96.
-Route of administration is required in the QUASAR study and in the TURANDOT study
-Line 23, “Finally” is not a correct word in this context
-Line 24, Delete “We explore”
-Line 26, Keep the same order as the manuscript, “anti-interleukin-23” should be before “anti-integrins” and “sphingosine-1-phosphate receptor modulators” in the end
-Line 67, Delete “Crohn´s Disease”
-Line 107, Define “CDAI” here.
-Lines 212 to 215 are repeated in lines 218 to 2020.
-Line 233 interleukin abbreviation should be defined before
-Line 328, “controlled” instead of “controlled”
-Lines 393, 406, and 493. Delete additional dots
-Line 557, Two points
-Line 393. Authors should describe the TOSCA study
-Line 558. Text in brackets no subscore>1 is not understood.
-Line 570, “between” instead of “netween”
-Line 599, “Minimal” should not be a capital letter
-Line 609. ”Iv” should be as “IV”
-Line 632. IFNg should be defined here
-Line 686, “Showed” is repeated
-Line 733 and 734. BMS and Cminss? Define them
-Lines 748 and 749 alpha4 and alpha4beta7 should be written in a symbol pattern as the rest of the text.
References:
-Reference 23 is needed in Table 1 in Guselkumab
-Reference number 49 does not match with text in line 357
-Reference number 81 is for TLR9 not for all the TLRs. Authors should delete it or find another review for the family of TLRs. I recommend the last one.
Figure
The lymphocyte name is missing in the figure
Yellow color is difficult to appreciate
Change order, “Peficitinib” before “Ritlecitinib”
Author Response
We thank the reviewer for the comments. We revised our manuscript as suggested.
Reviewer 2 Report
This manuscript depicted through reviewing of clinical trials with various emerging biodrugs. This list makes sense for readers interested with these fields. However, there are a lot of misspelling in abstract and tables.
And author should arranging future direction which these trials are going. Combination of biologics and head-to-head comparisons are one of future directions. Please give us further comments on this directions needed.
Thre are a lot of misspellings.
Author Response
We greatly appreciate the reviewer's advice. The manuscript has been revised by an English native speaker.
Reviewer 3 Report
Dear Editor
This is a well-written review article regarding new agents for managing inflammatory bowel diseases. Only minor comments were found.
#1. Page 2, Ontamalimab was mentioned as a drug name, and in the trial description, the product code was provided as PF-00547659. It would be better to use Ontamalimab as your description in the OPERA II study. Or consider the format as Obefazimod ( ABX464) section.
#2. Page 6, same as the previous comment.
#3.
Dear Editor
This is a well-written review article regarding new agents for managing inflammatory bowel diseases. Only minor comments were found.
#1. Page 2, Ontamalimab was mentioned as a drug name, and in the trial description, the product code was provided as PF-00547659. It would be better to use Ontamalimab as your description in the OPERA II study. Or consider the format as Obefazimod ( ABX464) section.
#2. Page 6, same as the previous comment.
Author Response
We greatly appreciated the reviewer’s comments. We revised our manuscript and made the suggested changes.
Round 2
Reviewer 2 Report
The revised manuscript has been re-written well according to reviewer's suggestions. It is acceptable as it's present form.